# Molecular Simulations of Sputtering Preparation and Transformation of Surface Properties of Au/Cu Alloy Coatings Under Different Incident Energies

**Linxing Zhang** [1], **Sen Tian** [2,3,*] and **Tiefeng Peng** [4,*]

[1] College of Aerospace Engineering, Chongqing University, Chongqing 400044, China; 201731131023@cqu.edu.cn
[2] State Key Laboratory of Coal Mine Disaster Dynamics and Control, Chongqing University, Chongqing 400044, China
[3] School of Civil Engineering, The University of Queensland, Brisbane, QLD 4072, Australia
[4] Key Laboratory of Ministry of Education for Solid Waste Treatment and Resource Recycle, Southwest University of Science and Technology, Mianyang 621010, China
* Correspondence: sentian@cqu.edu.cn (S.T.); pengtiefeng@cqu.edu.cn (T.P.); Tel.: +86-23-6510-5093 (S.T.); +86-816-6089453 (T.P.)

**Abstract:** The surface properties of coatings during deposition are strongly influenced by temperature, particle fluxes, and compositions. In addition, the precursor incident energy also affects the surface properties of coatings during sputtering. The atomistic processes associated with the microstructure of coatings and the surface morphological evolution during sputtering are difficult to observe. Thus, in the present study, molecular dynamics simulation was employed to investigate the surface properties of Au/Cu alloy coatings (Cu substrate sputtering by Au atoms) with different incident energies (0.15 eV, 0.3 eV, 0.6 eV). Subsequently, the sputtering depth of the Au atoms, the particle distribution of the Au/Cu alloy coating system, the radial distribution function of particles in the coatings, the mean square displacement of the Cu atoms in the substrate, and the roughness of the coatings were analyzed. Results showed that the crystal structure and the sputtering depth of Au atoms were hardly influenced by the incident energy, and the incident energy had little impact on the motion of deep-lying atoms in the substrate. However, higher incident energy resulted in higher surface temperature of coatings, and more Au atoms existed in the coherent interface. Moreover, it strengthened the motion of Cu atoms and reduced the surface roughness. Therefore, the crystal structure of coatings and the motions of deep-lying atoms in the substrate are not influenced by the incident energy. However, the increase in incident energy will enhance the combination of coatings and the base while optimizing the surface structure.

**Keywords:** alloy coatings; incident energy; surface properties; molecular dynamics; radial distribution function; mean square displacement; roughness

## 1. Introduction

The rapid development of processing and manufacturing industries has placed increasing demands on the safety performance of operation tools and components. This requires the use of protective coatings characterized by high hardness [1,2], high strength and toughness [3,4], heat resistance [5], and wear and corrosion resistance [6,7]. The materials used for thermal protection systems and engine hot section components in the aerospace field are also often subjected to high temperature, mechanical shock, corrosion, and other coupling effects [8]. The use of surface coating technology is an effective means of satisfying the above requirements. Temperature is an important

determinant of properties for coating materials during their preparation and service [9]. Guo et al. [10] conducted a cyclic thermal loading test on a 1.5 nm yttria-stabilized zirconia (YSZ) thermal barrier coating, which revealed that the substrate temperature seriously affected the failure characteristics of the coating. Through analyses of the thermal stress, coating thickness, and substrate temperature during thermal spray cooling, Song et al. [11] proposed a corresponding thermal stress analysis model and found that the heat transfer behavior between the coating and the substrate affected the thermal stress distribution. In addition, the surface properties of coatings seriously affect their service performance, a topic that has received extensive scholarly attention as well. However, given the tiny interface structure of coatings, studying their surface properties by conventional experiments and theories is challenging.

Molecular simulation [12–15] has gained wide application in the research of material properties in recent years with the fast advancement of computer technology. Molecular dynamics (MD) is an indispensable computational tool for studying the microstructure and surface morphological evolution of coatings and thin films upon varying incident flux compositions [16,17] and energies [18–20]. MD simulations allow the structure and properties of molecular systems to be analyzed via statistical thermodynamics [18–23]. Fu et al. [24] calculated the generalized stacking fault energy curve of vanadium nitride (VN) ceramic film in the possible slip direction based on MD. They found that the nucleation of dislocations and the movement of partial dislocations were the main mechanisms for plastic deformation in the initial stage. Landman et al. [25] carried out an MD simulation of the loading of Ni needles on the Au surface and showed that the Au surface and the substrate exhibited a range of plastic deformations with the loading of Ni needles. An MD-based study on the surface roughness of Al found a decrease in the surface roughness with increasing substrate temperature [26]. When the temperature gradient during the sputtering preparation of Cu/Au alloy coatings was investigated further [27], results showed that the coherent growth structures and defects enhanced the particle mobility in the Cu/Au interface layer, which might result in failure of the coatings under thermal stress. The above body of literature indicates that the surface properties of coatings are subject to multiple factors, such as substrate temperature, coating material, and preparation process, of which temperature is a key factor.

Existing common technologies of surface coating include sputtering, surfacing, electroplating, electroless plating, chemical vapor deposition, chemical bonding, shot peening, etc. For the sputtering process, both the incident energy and substrate temperature of target materials are closely associated with the temperature factor [28–30]. Based on MD, Muller [31] studied the effects of different particle incident energies on coatings in a 2D Lennard-Jones (LJ) system and found that the formation of coatings underwent courses of local heating up, local melting, atomic refactoring, and systematic recrystallization. An MD simulation of the sputtering process of Cu, Al, and Ni particles on the metal surface by Hsieh et al. [32] revealed that the particle type, cluster size, and incident energy produced varying effects on the coating surface. Lei et al. [33] investigated the Cu deposition process on the Au substrate by MD and found that the incident energy was greatly influential to the morphology of the Cu epitaxial layer. Lattice mismatch occurred in the interface layer due to the different lattice constants of Cu and Au, and partial amorphous particles were present in the interface layer during the sputtering process; this eventually cooled to form a face-centered cubic (FCC) lattice structure. The above body of literature suggests the significant influence of incident energy on the surface properties of coatings. Hence, the effects of different incident energies on the surface properties of coatings are further studied herein using the MD method.

## 2. Simulation Method

MD simulations solve the motion trajectories of all particles in a certain thermodynamic system according to Newton's second law by taking the atoms or molecules in the system as the research unit. The accuracy of MD calculations relies heavily on the interaction potential function between atoms. As the potential function of high-temperature ceramic materials still requires further optimization [24],

this work uses the potential function of classical embedded atomic method (EAM) to study the effects of Cu substrate sputtering by Au atoms with different incident energies on the surface properties of coatings.

*2.1. Model and Computational Method*

Developed by Daw et al. [34], the EAM potential is a potential function model used to calculate the interaction between metal atoms [35–37]. It is based on the density functional theory, which considers that the electron density near a given atom is the sum of the electron densities of that atom and other atoms around it. The specific form of the EAM potential is as follows:

$$\Phi = \frac{1}{2}\sum_{i \neq j} \varphi(r_{ij}) + \sum_{i} F(\rho_i) \tag{1}$$

The first term is an interatomic pair potential [38], where $F$ denotes an embedding function. $\rho_i$, which is the electron density at the position of atom $i$, is calculated by summation over the electron densities $\rho(r_{ij})$ that derive from neighboring atoms $j$ at a distance $r_{ij}$ from the atom $i$.

$$\rho_i = \sum_{j} \rho(r_{ij}) \tag{2}$$

$F$ and $\rho$ are functions of several parameters, which can be determined by fitting known experimental and density functional theory data, such as elastic constant, equilibrium lattice constant, bulk modulus, and vacancy formation energies. The proper description of surface properties also requires fitting surface energies, adatom, and admolecule adsorption energies and migration energies for different reaction pathways [21,22]. Although parameterized with respect to both bulk and surface properties [39], the prediction of reaction pathways and reaction rates obtained via molecular dynamics based on empirical models [40] need verification via ab initio MD [41]. Nevertheless, the EAM potential parameters adopted in this study can simulate the thermal and mechanical properties of Au/Cu binary system rather accurately and are therefore considered reliable [42].

*2.2. Materials and Preparation Process*

The computational model of the Au/Cu alloy coating sputtering preparation is demonstrated in Figure 1. The size (Å) of Cu substrate was $X$: 72, $Y$: 72, $Z$: 36 and consisted of three parts of 16,000 atoms. Its bottom was the lower boundary of the simulation box defined by the fixed Cu atoms. Other substrate atoms were first thermostated at 300 K for 1 ns in the *NVT* ensemble (constant number ($N$), volume ($V$), and temperature ($T$)). Then, the intermediate Cu particles were thermostated at around 300 K in the *NVT* ensemble as a cooling heat source for the sputtering process. Meanwhile, the top Cu particles were relaxed in the *NVE* ensemble (constant number ($N$), volume ($V$), and energy ($E$)) for 1 ns. During the sputtering preparation of coatings, this part of Cu particles acted directly with the sputtering Au atoms. The Au atoms were generated randomly from the top of the simulation box (250 Å apart from the substrate surface) and sputtered onto the bottom Cu substrate from top down at a rate of $7.5 \times 10^{24}$ atom/(s·cm$^2$) [43] under the *NVE* ensemble. A total of 4000 Au atoms were sputtered. Various incident energies (0.15 eV, 0.3 eV, and 0.6 eV) were set along the $Z$-axis to obtain three different coating materials by sputtering. Finally, the prepared system was annealed under *NVT* ensemble and cooled to 300 K to statistically analyze the surface properties of the coatings.

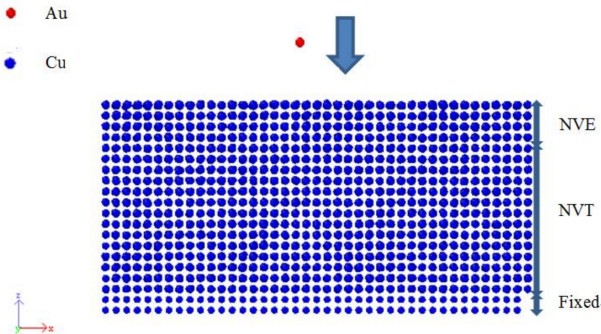

**Figure 1.** The system model of sputtering preparation for the Au/Cu alloy coatings.

The open-source LAMMPS software (Sandia National Laboratories, Albuquerque, NM, USA) was used in this study for the MD simulation [44,45]. The time step in the simulation was set to 1 fs, the *X* and *Y*-axis directions of the simulation box were periodic boundary conditions, while the *Z*-axis direction was the mirror boundary condition. The Langevin algorithm was employed to control the temperature of the system, whereas the velocity Verlet algorithm was used to solve the equations of motion for particles [44,46].

## 3. Results and Discussion

### 3.1. Particle Distribution

Au atoms were sputtered onto the surface of the substrate, which developed a coherent interface with the Cu atoms at the substrate and formed an FCC lattice structure in the vertical direction (*Z*-axis direction) after cooling [27,42,47]. The structural analysis and post-processing were performed with OVITO [27,42]. Figure 2 illustrates the atomic distribution of the coating system along the *Z*-axis direction, where the *Z*-axis origin is the bottommost end of the substrate.

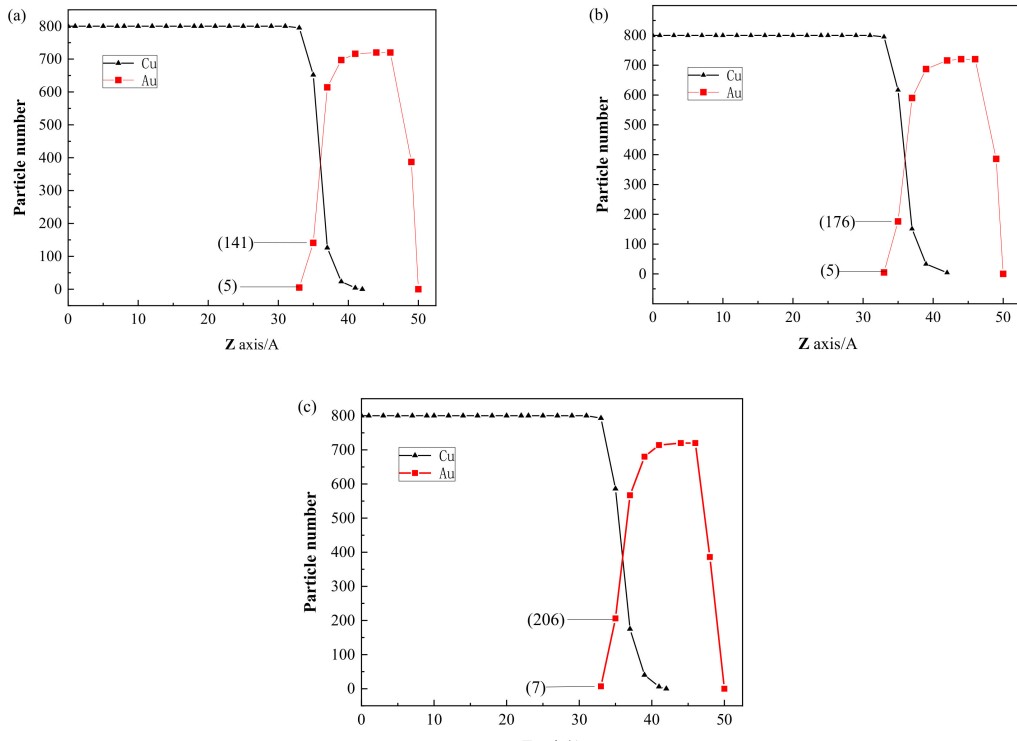

**Figure 2.** Particle distribution in *Z*-axis. (**a**) Incident energy of 0.15 eV; (**b**) incident energy of 0.3 eV; (**c**) incident energy of 0.6 eV.

As can be seen, the sputtering depth of Au atoms was about 33 Å all along at various incident energies, mainly because the maximum energy of incident Au atoms was insufficient to break down and replace the lower depth Cu atoms. The fundamental cause of this phenomenon was an increase in the kinetic energy of incident atoms with the increasing incident energy. Nevertheless, the atomic numbers of Au atoms with incident energies of 0.15 eV, 0.3 eV, and 0.6 eV were 5, 5, and 7, respectively, in the first layer of the coherent interface. The atomic numbers were 141, 176, and 206, respectively, in the second layer. After the highly energized Au atoms hit the substrate, the temperature of the Cu substrate in the *NVE* ensemble rose more drastically, which caused the melting of the substrate and eventually led to entry of more Au atoms to the deep layer of the substrate to facilitate the bonding of the coatings with the substrate.

Radial distribution function (RDF) describes the ratio of the probability of presence of another particle to the random distribution at a certain distance around a given particle, which is often used to detect the distribution of particles in a system and their corresponding state [48]. Its statistical formula is as follows:

$$G(r) = \frac{\mathrm{d}N}{\rho 4\pi r^2 \mathrm{d}r} \tag{3}$$

where $G(r)$ denotes the RDF of the corresponding particle at a radius $r$ or, in other words, the ratio of the regional density to the average density $\rho$ of system. $N$ denotes the number of particles. Figure 3 displays the RDF of the entire coating systems after annealing. At different incident energies, the trends and numerical values of the particle RDF curves were almost consistent. The first peaks were all located at 2.5 Å, after which a series of peaks appeared, indicating that the particles in the different systems were distributed orderly in both near and remote ranges and that the compositions of crystal structures were similar. The Au/Cu systems obtained by sputtering at three different incident energies basically showed no difference in the crystal structure composition at the coherent interface.

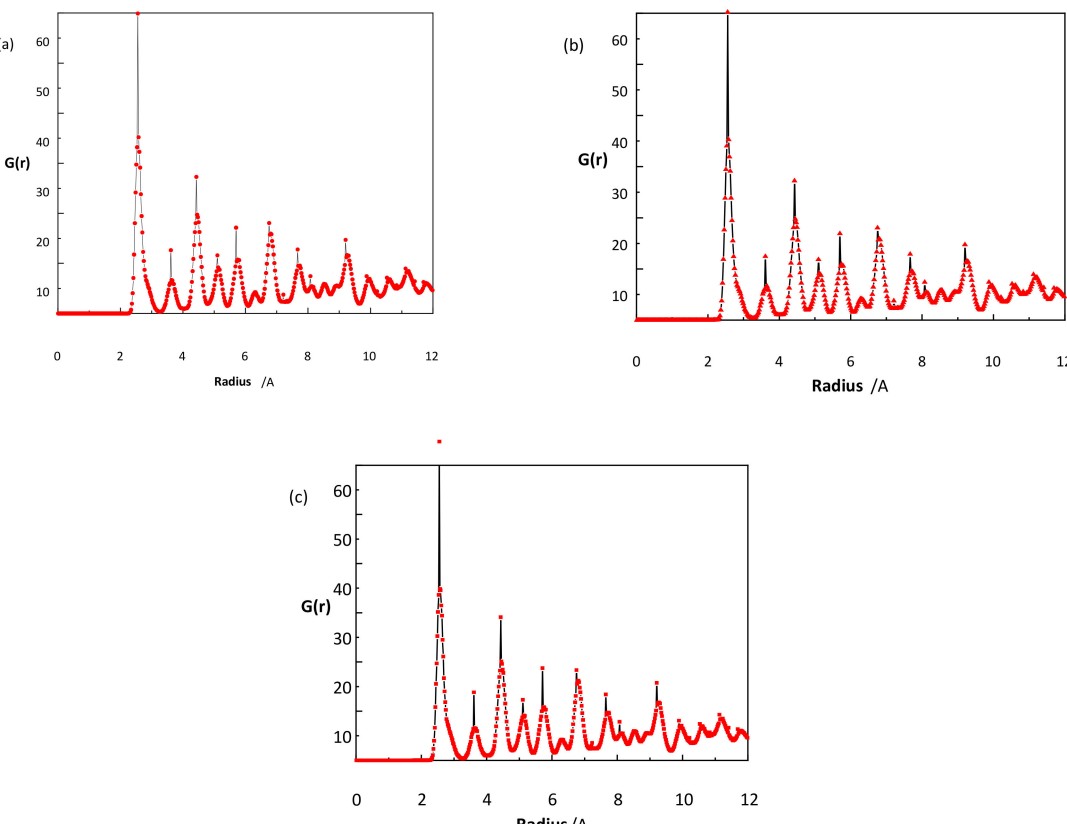

**Figure 3.** Radial distribution function of particles in coatings. (**a**) Incident energy of 0.15 eV; (**b**) incident energy of 0.3 eV; (**c**) incident energy of 0.6 eV.

### 3.2. Mean Square Displacement of Substrate Atoms

In MD simulation, the position of particles moves continuously with time, and each particle has varied position at each instant. Mean square displacement (MSD), which is the average of the square of particle displacement, is often used to analyze the mobility of particles [27]. Given the Cu atoms at the substrate were collided by Au atoms mainly in the sputtering phase to undergo large displacements, we analyzed the motion properties of substrate Cu atoms in two ensembles (*NVT* and *NVE*) during the sputtering process by MSD, as shown in Figure 4. In the *NVT* ensemble, the MSD of the Cu atoms was maintained at about 0.05 Å$^2$. This not only indicated that the Cu atoms of the *NVT* ensemble vibrated fixed in the vicinity of crystal lattice under the EAM potential function but also suggested that the incident energy of Au atoms was limited, which was unable to effectively influence the movement of this part of Cu atoms. In the *NVE* ensemble, on the other hand, the MSD of Cu atoms increased almost linearly in the sputtering phase with the increasing incident energy, which eventually leveled off. As larger incident energy means larger kinetic energy of Au atoms, after sputtering and impacting the substrate particles of the *NVE* ensemble, the kinetic energy acquired by Cu atoms increased, which led to enhanced mobility of Cu atoms in the system. As the number of sputtered Au particles reached 4000, the Au/Cu system relaxed and cooled by the action of the *NVT* ensemble, and the whole system underwent a recrystallization process. Upon completion of the recrystallization, the Cu particles at the surface layer of the coating (*NVE* ensemble) vibrated at the lattice position, and their MSDs were stabilized at certain values.

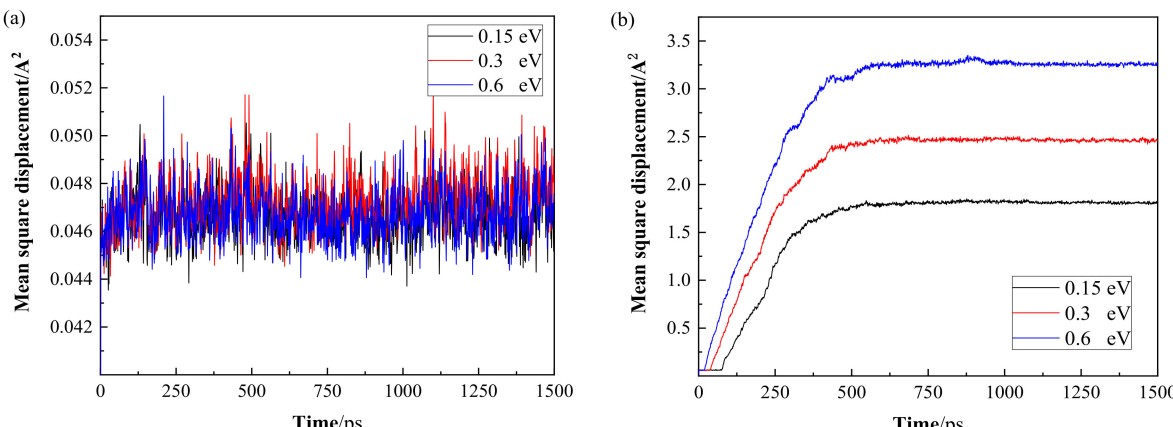

**Figure 4.** Mean square displacement (MSD) of substrate Cu atoms. (**a**) Cu atoms in *NVE* ensemble; (**b**) Cu atoms in the *NVT* (constant number (*N*), volume (*V*), and temperature (*T*)) ensemble.

### 3.3. Surface Roughness of Coatings

Surface roughness, as a major indicator of coatings, is inevitably varied by the incident energy. In this study, the roughness *R* was calculated in terms of the root mean square of the height position of coating surface particles [49–51]. Its statistical formula is as follows [52]:

$$R = \sqrt{\frac{\sum\limits_{i=1}^{n} \left(Z_i - \overline{Z}\right)^2}{n}} \tag{4}$$

where *n* denotes the total number of particles on the coating surface, $Z_i$ denotes the vertical coordinate of surface particles, and $\overline{Z}$ denotes the mean vertical coordinate thereof. The surface roughness of the Au/Cu alloy coatings with the incident energy of 0.15 eV, 0.3 eV, and 0.6 eV before and after annealing was obtained through calculations, as shown in Table 1. As can be seen, there were certain changes in the surface roughness of coatings before and after annealing. Both before and after annealing, the surface roughness decreased with the increasing incident energy. At the same incident energies,

the roughness values after annealing were all less than that before annealing. This was because the larger incident energy raised the temperature at the coating surface to make its state tending to a molten liquid phase, and the surface roughness was reduced correspondingly under surface tension. Meanwhile, during the annealing process, the structure of the coating systems was optimized to reduce the specific area and lower the free energy, and their surface roughness was also reduced further.

**Table 1.** Surface roughness of Au/Cu alloy coatings before and after annealing.

| Status of Coatings | Incident Energy | | |
|---|---|---|---|
| | 0.15 eV | 0.3 eV | 0.6 eV |
| Before annealing (Å) | 2.19 | 2.05 | 2.03 |
| After annealing (Å) | 2.05 | 1.93 | 1.87 |

## 4. Conclusions

In this study, the surface properties of Au/Cu alloy coatings prepared by sputtering at different incident energies were studied by analyzing the particle distribution, particle radial distribution function, particle motion mean square displacement, and roughness of the coatings by the MD simulation method. Under the simulated conditions of this study, the incident energy produced no effect on the sputtering depth of Au atoms. Nevertheless, greater incident energy caused more drastic rise in the substrate temperature, so more Au atoms were mixed with the substrate Cu atoms to form a coherent interface, which facilitated the bonding of the coatings with the substrate. Incident energy had no substantial effect on the crystal structure of particles in various regions of the coatings, nor could it affect the motion of deep-lying atoms in the substrate. In spite of this, greater incident energy still signified stronger mobility of the Cu atoms in the surface layer of the substrate. The incident energy significantly influenced the surface roughness of coatings, as indicated by lower coating roughness with its increase. In addition, annealed coatings exhibited better roughness than those before annealing optimization.

**Author Contributions:** L.Z. participated in the design of the work, methodology, data interpretation, and analysis for the work; carried out the statistical analyses; and drafted the manuscript. S.T. designed the study; participated in data interpretation, analysis for the work, and methodology; carried out the statistical analyses; and drafted the manuscript. T.P. participated in design of the work, data interpretation, and analysis for the work and helped carry out the statistical analyses.

**Funding:** This study was financially supported by the Chongqing Research Program of Basic Research and Frontier Technology (cstc2018jcyjAX0522), the National Natural Science Foundation of China (11332013, 51502311), the Program for Changjiang Scholars and Innovative Research Team in University (IRT_17R112), and the Innovation Support Program for Chongqing Overseas Returnees (cx2018071).

**Acknowledgments:** The authors would like to acknowledge the colleagues from the State Key Laboratory of Coal Mine Disaster Dynamics and Control for their perspectives and suggestions related to data collection and statistical analysis.

**Conflicts of Interest:** The authors declare no conflict of interest.

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
