# Peer review of "Molecular Simulations of Sputtering Preparation and Transformation of Surface Properties of Au/Cu Alloy Coatings Under Different Incident Energies"

_metals, doi:10.3390/met9020259_

Round 1

Reviewer 1 Report

This ms presents a reasonably good molecular dynamics study of Cu surface reactions and surface structural evolution upon Au deposition using different Au energies. The ms is reasonably well written and presented. However, its quality needs to be improved. It is not acceptable in the present form. I provide below several detailed comments. I shall reconsider after the authors have amended all listed issues.

-------------------

Title

There is a typo in the title. Please correct “Proerties”!!

-------------------

Abstract

The first part of the abstract needs to be improved. What do the authors mean with “ingredient”? That does not sound correct in the context of coatings and sputtering. Also “size” seems inaccurate. Do the authors mean “thickness”

“As an important characteristic of coatings, the surface properties are influenced by temperature, ingredient, size, etc. Besides, the incident energy also affects the surface properties of coatings during the sputtering process. The coating’s own micro-structure and the sputtering are difficult to observe…”

I recommend rephrasing the part above as:

“The surface properties of coatings during deposition are strongly influenced by temperature, particle fluxes and compositions. Besides, the precursor incident energy also affects the surface properties of coatings during sputtering. The atomistic processes associated with coating’s microstructure and surface morphological evolution during sputtering are difficult to observe…”

-------------------

Introduction

1) The introduction starts with “The rapid development of processing and manufacturing industries has placed ever heightening demands on the material safety performance, such as high hardness, high strength, heat resistance, wear resistance and corrosion resistance.”

It is not clear the meaning of “materials” here. I assume that the authors intend to say “coating materials”. I advice to slightly rephrase and to add references:

“The rapid development of processing and manufacturing industries has placed ever heightening demands on the safety performance of operation tools and components.  This requires the use of protective coatings characterized by high hardness [1, 2], high strength and toughness [3, 4], heat resistance [5], wear and corrosion resistance [6, 7].”

2) The survey of the field for MD simulations should be expanded to include other studies specifically addressing the changes in surface morphology of hard coatings induced by different vapor compositions and particle energies. It also suggest rephrasing.

Molecular dynamics (MD) is an indispensable computational tool for studying the microstructure and surface morphological evolution of coatings and thin films upon varying incident flux compositions [8, 9] and energies [10-12]. MD simulations allow analyzing the structure and properties of molecular systems via statistical thermodynamics [10-14].

-------------------

Methods

1) The description of the EAM model is not sufficiently good. The authors should be more accurate. For example: “F denotes a mosaic function”. What is the meaning? Isnt’ this the embedding function? Is the “first term” a force or energy? If it is a pair interaction term, shouldn’t it have indexes i and j? Is the embedding function F a force or an energy? Please note that the expression should be consistent. If you have force (or energy) on the left-hand side, you should have a force (or energy) on the right-hand side as well.

Specify the meaning of rij , ρ(rij), and ρ_i. It is vague to say that “ρ(rij)” and “ρ_i” denote an electron density function. The description would read better as:

“ρ_i, which is the electron density at the position of atom i, is calculated by summation over the electron densities ρ(rij) that derive from neighbor atoms j at a distance rij from the atom i.”  

2) It is not fully clear from which paper the Cu and Au parameters were taken. After Eq. (2), the ms reads: “The EAM potential parameters adopted in this study can simulate the thermal and mechanical properties of Au/Cu binary system rather accurately [31].” Are the original parameters of Au and Cu taken from Ref. 31? Please make this clear.

3) The authors specify the ensembles used to model Cu atoms. They do not specify, however, the ensemble used for Au atoms. Since these are deposited at constant energy, the ensemble used for Au atoms must be NVE. Please specify in the text.

4) Related to the previous comment. Are all Au particles coming down orthogonally to the surface? Different incident angles? I do not see it specified. Please specify it in the text.

5) Regarding the reliability of the model used, the text reads: “F and ρ contain multiple specific parameters, which can be determined by fitting the known experimental data, such as elastic constant, equilibrium lattice constant, bulk modulus and vacancy formation energy. The EAM potential parameters adopted in this study can simulate the thermal and mechanical properties of Au/Cu binary system rather accurately [31].”

I recommend rephrasing and expanding this part to inform readers of the limited reliability of classical models. Also the description of F and ρ should be improved.

F and ρ are functions of several parameters, which can be determined by fitting known experimental and density functional theory data, such as elastic constant, equilibrium lattice constant, bulk modulus, and vacancy formation energies. Proper description of surface properties also requires fitting surface energies, adatom and admolecule adsorption energies and migration energies for different reaction pathways [13, 14]. Although parameterized with respect to both bulk and surface properties [15], the prediction of reaction pathways and reaction rates obtained via molecular dynamics based on empirical models [16] need verification via ab initio MD [17]. Nevertheless, the EAM potential parameters adopted in this study can simulate the thermal and mechanical properties of Au/Cu binary system rather accurately and are therefore considered reliable [31].”

-------------------

Results

1) Can the authors comment on choice of the incident energies used? They seem low… but perhaps sufficiently high considering that Cu is a relatively soft substrate.

Is the relative penetration of Au atoms in the Cu substrate as a function of Au energy (Fig. 2) considered significant? Do they consider the differences significant? Do the authors deem that the use of energies of the order of eV would have destroyed the substrate?

2) The top panel of Figure 4 (I mean Fig. 4(a)) is illegible. It could be removed and replaced by a comment in the text. If the authors intend to keep it, they should shrink the MSD scale from ~0.04 to ~0.055 A^2.

3) In table 1, the authors show roughness values. What are the units? Please specify.

Additional references

[1] T. Reeswinkel et al. Structure and mechanical properties of TiAlN-WNx thin films, Surface and Coatings Technology 205 (2011) 4821.

[2] M. Mikula et al. Toughness enhancement in highly NbN-alloyed Ti-Al-N hard coatings, Acta Materialia 121 (2016) 59-67.

[3] H. Kindlund et al. Effect of WN content on toughness enhancement in V1-xWxN/MgO(001) thin films, Journal of Vacuum Science & Technology A 32 (2014) 030603.

[4] M. Mikula et al. Experimental and computational studies on toughness enhancement in Ti-Al-Ta-N quaternaries, Journal of Vacuum Science & Technology A: Vacuum, Surfaces, and Films 35(6) (2017) 060602.

[5] P.H. Mayrhofer, C. Mitterer, L. Hultman, H. Clemens, Microstructural design of hard coatings, Progress in Materials Science 51(8) (2006) 1032-1114.

[6] A.A. Voevodin, J.S. Zabinski, Supertough wear-resistant coatings with 'chameleon' surface adaptation, Thin Solid Films 370(1-2) (2000) 223-231.

[7] Q. Yang, L.R. Zhao, P.C. Patnaik, X.T. Zeng, Wear resistant TiMoN coatings deposited by magnetron sputtering, Wear 261(2) (2006) 119-125.

[8] D. Edstrom, D.G. Sangiovanni, L. Hultman, I. Petrov, J.E. Greene, V. Chirita, Large-scale molecular dynamics simulations of TiN/TiN(001) epitaxial film growth, Journal of Vacuum Science & Technology A 34(4) (2016) 9.

[9] Z. Xu, Q. Zeng, L. Yuan, Y. Qin, M. Chen, D. Shan, Molecular dynamics study of the interactions of incident N or Ti atoms with the TiN(001) surface, Applied Surface Science 360 (2016) 946-952.

[10] D. Edström, D.G. Sangiovanni, L. Hultman, I. Petrov, J.E. Greene, V. Chirita, Effects of incident N atom kinetic energy on TiN/TiN (001) film growth dynamics: A molecular dynamics investigation, Journal of Applied Physics 121(2) (2017) 025302.

[11] D. Adamovic, E.P. Munger, V. Chirita, L. Hultman, J.E. Greene, Low-energy ion irradiation during film growth: Kinetic pathways leading to enhanced adatom migration rates, Applied Physics Letters 86(21) (2005) 211915.

[12] D. Adamovic, V. Chirita, E.P. Munger, L. Hultman, J.E. Greene, Enhanced intra- and interlayer mass transport on Pt(111) via 5-50 eV Pt atom impacts on two-dimensional Pt clusters, Thin Solid Films 515(4) (2006) 2235-2243.

[13] D.G. Sangiovanni, F. Tasnadi, L. Hultman, I. Petrov, J.E. Greene, V. Chirita, N and Ti adatom dynamics on stoichiometric polar TiN(111) surfaces, Surface Science 649 (2016) 72-79.

[14] D.G. Sangiovanni, D. Edström, L. Hultman, I. Petrov, J.E. Greene, V. Chirita, Ti adatom diffusion on TiN(001): Ab initio and classical molecular dynamics simulations, Surface Science 627 (2014) 34.

[15] D.G. Sangiovanni, D. Edström, L. Hultman, V. Chirita, I. Petrov, J.E. Greene, Dynamics of Ti, N, and TiNx (x = 1–3) admolecule transport on TiN(001) surfaces, Physical Review B 86 (2012) 155443.

[16] D. Edstrom, D.G. Sangiovanni, L. Hultman, V. Chirita, I. Petrov, J.E. Greene, Ti and N adatom descent pathways to the terrace from atop two-dimensional TiN/TiN(001) islands, Thin Solid Films 558 (2014) 37.

[17] D.G. Sangiovanni, A.B. Mei, D. Edström, L. Hultman, V. Chirita, I. Petrov, J.E. Greene, Effects of surface vibrations on interlayer mass transport: Ab initio molecular dynamics investigation of Ti adatom descent pathways and rates from TiN/TiN (001) islands, Physical Review B 97 (2018) 035406.

Author Response

Response to Reviewer 1 Comments

Dear Editors and Reviewers,

 Thank you for your letter and for the reviewers’ comments concerning our manuscript entitled“Molecular Simulations of Sputtering Preparation and Surface Properties transformation of Au/Cu Alloy Coatings” (metals-435648). Those comments are all valuable and very helpful for revising and improving our paper, as well as the important guiding significance to our researches. We have studied comments carefully and have made correction which we hope meet with approval. Revised portion are marked in red in the paper. The main corrections in the paper and the response to the reviewer’s comments are as flowing:

Point 1: There is a typo in the title. Please correct “Proerties”!! 

Response 1: Done. We are very sorry for our negligence of this spelling error. We have checked the all Propertiesin the revision.

Point 2: “As an important characteristic of coatings, the surface properties are influenced by temperature, ingredient, size, etc. Besides, the incident energy also affects the surface properties of coatings during the sputtering process. The coating’s own micro-structure and the sputtering are difficult to observe…” . I recommend rephrasing the part above as: “The surface properties of coatings during deposition are strongly influenced by temperature, particle fluxes and compositions. Besides, the precursor incident energy also affects the surface properties of coatings during sputtering. The atomistic processes associated with coating’s microstructure and surface morphological evolution during sputtering are difficult to observe…”

Response 2: Done. We have re-written this part in the Abstract section according to the your suggestion. Please see the revision (in page 1, line 14-18).

Point 3: It is not clear the meaning of “materials”  in the “Introduction”. I assume that the authors intend to say “coating materials”. I advice to slightly rephrase and to add references: “The rapid development of processing and manufacturing industries has placed ever heightening demands on the safety performance of operation tools and components.  This requires the use of protective coatings characterized by high hardness [1, 2], high strength and toughness [3, 4], heat resistance [5], wear and corrosion resistance [6, 7].”

Response 3: Done. We have re-written this part, and added the recommended references. Please see the revision (in page 1, line 34-38). 

Point 4: The survey of the field for MD simulations should be expanded to include other studies specifically addressing the changes in surface morphology of hard coatings induced by different vapor compositions and particle energies. It also suggest rephrasing. Molecular dynamics (MD) is an indispensable computational tool for studying the microstructure and surface morphological evolution of coatings and thin films upon varying incident flux compositions [8, 9] and energies [10-12]. MD simulations allow analyzing the structure and properties of molecular systems via statistical thermodynamics [10-14].

Response 4: Done. We have made corrections according to the reviewer’s suggestion, and  the  recommended references have been added. Please see the revision (in page 2, line 53-57). 

Point 5: The description of the EAM model is not sufficiently good. The authors should be more accurate. For example: “F denotes a mosaic function”. What is the meaning? Isnt’ this the embedding function? Is the “first term” a force or energy? If it is a pair interaction term, shouldn’t it have indexes i and j? Is the embedding function F a force or an energy? Please note that the expression should be consistent. If you have force (or energy) on the left-hand side, you should have a force (or energy) on the right-hand side as well.

Specify the meaning of rij , ρ(rij), and ρ_i. It is vague to say that “ρ(rij)” and “ρ_i” denote an electron density function.

Response 5:  We revised the associated expressions and descriptions in section 2.1 in the revision. Thank you for pointing this out, and we hope that this revision presents the expressions and descriptions more clearly and consistent.  

Then the  description has been corrected as: “ρ_i, which is the electron density at the position of atom i, is calculated by summation over the electron densities ρ(rij) that derive from neighbor atoms j at a distance rij from the atom i.”   

Please see the revision (in page 3, line 99-101). 

Point 6: It is not fully clear from which paper the Cu and Au parameters were taken. After Eq. (2), the ms reads: “The EAM potential parameters adopted in this study can simulate the thermal and mechanical properties of Au/Cu binary system rather accurately [31].” Are the original parameters of Au and Cu taken from Ref. 31? Please make this clear.

Response 6: The parameters of Au and Cu are taken from our previous works, Ref [27] and [42] in the revision. Also, we tested the thermal and mechanical properties of the present parameters in Ref. [42].

Point 7: The authors specify the ensembles used to model Cu atoms. They do not specify, however, the ensemble used for Au atoms. Since these are deposited at constant energy, the ensemble used for Au atoms must be NVE. Please specify in the text.

Response 7: Done. Thank you for pointing this out. The Au atoms were modeled in NVE ensemble. Noted the NVE ensemble used here is to update the position and velocity of Au atoms. ONLY the incident energy in Z axis of Au atoms is maintained at 0.15 eV, 0.3 eV and 0.6 eV, respectively. We had specified it in the revision.

Point 8: Related to the previous comment. Are all Au particles coming down orthogonally to the surface? Different incident angles? I do not see it specified. Please specify it in the text.

Response 8: As aforementioned, ONLY the incident energy in Z axis of Au atoms is maintained at 0.15 eV, 0.3 eV and 0.6 eV, respectively. The kinetic energy in X and Y directions of Au atoms are controlled by NVE ensemble of LAMMPS. Again, noted the NVE ensemble employed here is to update the position and velocity of Au atoms. Thus, the incident angles are random.

Point 9:  Regarding the reliability of the model used, the text reads: “F and ρ contain multiple specific parameters, which can be determined by fitting the known experimental data, such as elastic constant, equilibrium lattice constant, bulk modulus and vacancy formation energy. The EAM potential parameters adopted in this study can simulate the thermal and mechanical properties of Au/Cu binary system rather accurately [31].”

I recommend rephrasing and expanding this part to inform readers of the limited reliability of classical models. Also the description of F and ρ should be improved.

Response 9: Done. Page 3, line 102-111, F and ρ are functions of several parameters, which can be determined by fitting known experimental and density functional theory data, such as elastic constant, equilibrium lattice constant, bulk modulus, and vacancy formation energies. Proper description of surface properties also requires fitting surface energies, adatom and admolecule adsorption energies and migration energies for different reaction pathways [13, 14]. Although parameterized with respect to both bulk and surface properties [15], the prediction of reaction pathways and reaction rates obtained via molecular dynamics based on empirical models [16] need verification via ab initio MD [17]. Nevertheless, the EAM potential parameters adopted in this study can simulate the thermal and mechanical properties of Au/Cu binary system rather accurately and are therefore considered reliable [31].” was corrected according to the comment, and all recommended references has been added. Thank you for the suggestion.

Point 10:  Can the authors comment on choice of the incident energies used? They seem low… but perhaps sufficiently high considering that Cu is a relatively soft substrate.  Is the relative penetration of Au atoms in the Cu substrate as a function of Au energy (Fig. 2) considered significant? Do they consider the differences significant? Do the authors deem that the use of energies of the order of eV would have destroyed the substrate?

Response 10: The value of incident energies are chosen to produce Au/Cu thin films, which we investigated in our previous work Ref [27]. The present results indicate that there is no such function relationship between the penetration Au atoms and incident energy. And the Au atoms will destroy the substrate if the incident energy is high enough.

Point 11:  The top panel of Figure 4 (I mean Fig. 4(a)) is illegible. It could be removed and replaced by a comment in the text. If the authors intend to keep it, they should shrink the MSD scale from ~0.04 to ~0.055 A^2.

Response 11: Done. We have revised the Fig. 4(a), shrinked the MSD scale from ~0.04 to ~0.055 A^2 as is suggested.  

Point 12:  In table 1, the authors show roughness values. What are the units? Please specify.

Response 12: Done. We have  specified the units of the roughness values in Table 1. 

   We appreciate for Editors and Reviewers’ warm work earnestly, and hope that the correction will meet with approval. Once again, thank you very much for your comments and suggestions.

With kind regards,

Sen Tian 

Reviewer 2 Report

The title itself has spelling error “Surface Proerties”. There are various places in paper that shows poor sentence structure and grammar. The paper must be carefully proof-read for spelling mistake and English language before re-submission

1.      Among parameters investigated quenching rate and heat of mixing of components are key parameters for the deposition of metallic alloys.  Morphology, texture, surface characteristics, and crystallinity are affected with variation of mixing of component and rate of deposition. Also, deposition parameters will define the grain size of phase(s) deposited for various metal on the substrate.

2.      Materials in sub-section 2.2 is suggestive of some experimental work done, which is not the case need to change the sub heading.

3.      NVT and NVE is not defined anywhere in paper. In molecular simulation NVE: constant number (N), volume (V), and energy (E); the sum of kinetic (KE) and potential energy (PE) and NVT: constant number (N), volume (V), and temperature (T); T is regulated via a thermostat (assuming this is what author means). As a practice these should always be defined for readers to understand.

4.      In line 126 “which will develop a coherent interface with the Cu atoms at the substrate and form an FCC lattice structure in the vertical direction (Z-axis)”. If this is standard outcome, then reference must be provided if this is what authors are expecting then sentence needs to be re-written.

5.      Author briefly mention incident energy of Au is insufficient to breakdown Cu atoms but do not discuss what is its effects at interface. Also, in line 136 authors discussed the effect of thehighly energized Au atoms, which may cause the melting of Cu substrate and leads to entry of more Au atoms. What is happening at interface is not known/clear. Due to melting is there any possibility of various phases of coating on Cu surface. Such as, Au on copper surface and Au-Cu on Cu surface. Also, what would be possible causes of formation of void surfaces where no Au is present. Fig.2 illustrates the atomic distribution of the coating system which shows equal distribution in all three incident energies. In line 136 what is significance of melting?  In Cu melt phase is there a possibility of Au being dominated by copper such that Au penetrated and on topical surface is it more Cu and less Au or Cu is completely displaced. Structure evolution during a continuous heating/melting under various incident energies is not clearly written.

6.      It is not very clear if rate of deposition can be an important factor for deposition of coatings. Also upon cooling of Au/Cu system when system undergoes restructuring is there possibility of any amorphous phase. In line 176 authors discuss the reformation upon cooling. Is there formation of crystalline intermetallic compounds or amorphous alloys may exist. Are there various parameters for existence of any such phases. Does the incident energy impact existence of such phases?

7.       Increase in temperature with increasing incident energies should affect the crystallite size of Au/Cu due to increased penetration and higher diffusivity of Au. The reduced hardness at higher incident energy may be due to formation of amorphous phases. Author needs to investigate the crystallinity of coatings before assuming the change of surface roughness at higher incident energy.

Author Response

Response to Reviewer 2 Comments

Dear Editors and Reviewers,

The comments from the Reviewer 2 have been carefully concerned and addressed in the revision. The revised parts are marked as red color in the revision. Please find the comments and corresponding response below.

Point 1: The title itself has spelling error “Surface Proerties”. There are various places in paper that shows poor sentence structure and grammar. The paper must be carefully proof-read for spelling mistake and English language before re-submission .

Response 1: Done. It is our negligence and we are sorry about this. We have checked the all properties in the revision.  The English in the revision has been checked carefully, and we have had the manuscript polished with a professional assistance in English writing.

Point 2: Among parameters investigated quenching rate and heat of mixing of components are key parameters for the deposition of metallic alloys. Morphology, texture, surface characteristics, and crystallinity are affected with variation of mixing of component and rate of deposition. Also, deposition parameters will define the grain size of phase(s) deposited for various metal on the substrate.

Response 3: Done.  The deposition parameters have been addressed in the revision.

Point 3:  Materials in sub-section 2.2 is suggestive of some experimental work done, which is not the case need to change the sub heading.

Response 3: Done. We have made correction on sub heading 2.2 according to the Reviewer’s comments. 

Point 4: NVT and NVE is not defined anywhere in paper. In molecular simulation NVE: constant number (N), volume (V), and energy (E); the sum of kinetic (KE) and potential energy (PE) and NVT: constant number (N), volume (V), and temperature (T); T is regulated via a thermostat (assuming this is what author means). As a practice these should always be defined for readers to understand.

Response 4: Done. We have re-written this part according to the Reviewer’s suggestion,  please see the revision (in page 3, line 116-119). Thank you for picking them up.

Point 5: In line 126 “which will develop a coherent interface with the Cu atoms at the substrate and form an FCC lattice structure in the vertical direction (Z-axis)”. If this is standard outcome, then reference must be provided if this is what authors are expecting then sentence needs to be re-written.

Response 5: In line 126 “which will develop a coherent interface with the Cu atoms at the substrate and form an FCC lattice structure in the vertical direction (Z-axis)”, this outcome or experimental phenomena is taken from our previous works. We have added the relevant references. Please see them in the revision (in page 4, line 136-139).

Point 6: Author briefly mention incident energy of Au is insufficient to breakdown Cu atoms but do not discuss what is its effects at interface. Also, in line 136 authors discussed the effect of thehighly energized Au atoms, which may cause the melting of Cu substrate and leads to entry of more Au atoms. What is happening at interface is not known/clear. Due to melting is there any possibility of various phases of coating on Cu surface. Such as, Au on copper surface and Au-Cu on Cu surface. Also, what would be possible causes of formation of void surfaces where no Au is present. Fig.2 illustrates the atomic distribution of the coating system which shows equal distribution in all three incident energies. In line 136 what is significance of melting?  In Cu melt phase is there a possibility of Au being dominated by copper such that Au penetrated and on topical surface is it more Cu and less Au or Cu is completely displaced. Structure evolution during a continuous heating/melting under various incident energies is not clearly written.

Response 6: As we described in the manuscript, the temperature of the substrate increase as the Au atoms impact on it. Since the structure evolution rapidly during the deposition, it is hard to discuss the continuous change of system. Thus, we discuss the parameters of system before and after the deposition. Thank you for the suggestion.

Point 7:  It is not very clear if rate of deposition can be an important factor for deposition of coatings. Also upon cooling of Au/Cu system when system undergoes restructuring is there possibility of any amorphous phase. In line 176 authors discuss the reformation upon cooling. Is there formation of crystalline intermetallic compounds or amorphous alloys may exist. Are there various parameters for existence of any such phases. Does the incident energy impact existence of such phases?

Response 7: The rate of deposition is another important factor for coatings. Here, the rate of deposition is 7.5 × 1024 atom/s·cm2 which is taken from Ref. [43] in the revision. And there will be amorphous phase in the system when cooling. More details had been discussed in our previous works(Qibin Li, Cheng Huang, Yunpei Liang*, Tao Fu, Tiefeng Peng*. Molecular dynamics simulation of nanoindentation of Cu/Au thin films at different temperatures. Journal of Nanomaterials, 2016, 2016: 9265948.;Qibin Li*, Xianghe Peng, Tiefeng Peng*, Qizhong Tang, Chao Liu, Xiaoyang Shi. Molecular dynamics simulations of coating process: influences of thermostat methods. Journal of Computational and Theoretical Nanoscience, 2016, 13: 4629-4633.;Qibin Li*, Xianghe Peng, Tiefeng Peng*, Qizhong Tang, Xiaomin Zhang, Cheng Huang. Molecular dynamics simulation of Cu/Au thin films under temperature gradient. Applied Surface Science, 2015, 357: 1823-1829.).

Point 8: Increase in temperature with increasing incident energies should affect the crystallite size of Au/Cu due to increased penetration and higher diffusivity of Au. The reduced hardness at higher incident energy may be due to formation of amorphous phases. Author needs to investigate the crystallinity of coatings before assuming the change of surface roughness at higher incident energy.

Response 8: As we answered in question (3), the crystallinity of coatings had been discussed in our previous works. The present calculation of surface roughness had also used by other researchers Ref [49-52] in the revision.

We tried our best to improve the manuscript and made some changes in the manuscript, and hope that the correction will meet with approval. Thank you very much for your comments and suggestions.

With kind regards,

Sen Tian 

Reviewer 3 Report

In the reviewer's opinion, the submitted work does not meet the criteria of the article. The work contains only four figures, one small table and a short text that does not exhaust the issues raised. This type of work suggests a Communication. It is suggested to reject the submitted manuscript in its present form.

Author Response

Response to Reviewer 3 Comments 

Dear Editors and Reviewers,

Thank you and your reviewers for the extremely helpful comments provided for our paper. Please find the “comments and corresponding response” below.

Point 1: In the reviewer's opinion, the submitted work does not meet the criteria of the article. The work contains only four figures, one small table and a short text that does not exhaust the issues raised. This type of work suggests a Communication.

Response 1:

Thank you very much for your comments. First, the atomistic processes associated with coating’s microstructure and surface morphological evolution during sputtering are difficult to observe. On the basis of this main purpose, the molecular dynamics simulation is employed to investigate the coating surface properties (Cu substrate sputtering by Au atoms) with different values of incident energies (0.15eV, 0.3eV, 0.6eV) in this paper. The Au atoms’ incident depth, particle distribution of coating system, radial distribution function of particles in coating, mean square displacement of the base and roughness of coatings are analyzed. The above presents the research object and method. Molecular Dynamics Simulations are carried out and some important conclusions are obtained in this paper. The results show that the crystal structure and the sputtering depth of Au atoms are hardly influenced by the incident energy. And the energy of incident atoms impacts little on the bottom of base. And the higher incident energy will result in higher surface temperature of coating and more Au atoms exist in the coherent layer. The above are also the main innovation spots of this study. 

Addionally, this study is based on our previous study, and the method had been used for distinguishing the crystal structure in our previous works. Although the data analysis, including Figures and Table, on this seems somewhat limited, the present work focus on the surface effect caused by the incident energy, and the study has a comparative basis for the independence and integrity of the research purpose.

  We would like to discuss further microcosmic surface properties, such as the degree of order/disorder in the crystal structure in our future works. Many corrections have been made in the revision, we hope that the revision addresses your concerns. Thank you for your help.

  With kind regards,

  Sen Tian 

Reviewer 4 Report

The manuscript by Zhang et al. describes application of MD simulation in coatings surface properties during the sputtering process. The paper is clearly written and the conclusions are supported by the results. I recommend publication of this paper subject to the following minor revisions:

1-Line 122: The Velocity-Verlet algorithm needs a reference.

2-Line 123: “motion equation of particles” à equations of motion for particles

3-On page 4, lines 126, 127: How do the authors confirm that Au atoms sputtered on the Cu form an FCC lattice? There are order parameters in the literature for accurate determination of crystal structure (see for example Phys. Chem. Chem. Phys. 2018, 20, 27059).

4-On page 5, please define the parameters in the equation for the radial distribution function.

5-Page 7, regarding the surface roughness, the results show that changes in the surface roughness occurs upon sputtering. Is it possible to quantify the degree of order/disorder in the crystal structure before and after sputtering? Order parameters can be of help in this case.

Author Response

Response to Reviewer 4 Comments

Dear Editors and Reviewers,

Thank you for reviewing our paper and providing useful comments. The comments from the reviewer have been carefully concerned and addressed in the revision. The revised parts are marked as red color in the revision. Please find the “comments and corresponding response” below.The comments from the Reviewer 2 have been carefully concerned and addressed in the revision. The revised parts are marked as red color in the revision. Below is our point-by-point response to the referee’s comments.

Point 1: Line 122: The Velocity-Verlet algorithm needs a reference.

Response 1: Done.  Relevant references have been added, please see them in the revision (in page 4, line 131-133)

Point 2: Line 123: “motion equation of particles” à equations of motion for particles.

Response 2: Done. These have been corrected. Thank you for picking them up.

Point 3: On page 4, lines 126, 127: How do the authors confirm that Au atoms sputtered on the Cu form an FCC lattice? There are order parameters in the literature for accurate determination of crystal structure (see for example Phys. Chem. Chem. Phys. 2018, 20, 27059).

Response 3: Done. Thank you very much for you suggestion. The FCC lattice is analyzed by OVITO software, the structural analysis and post-processing were performed with OVITO [27, 42]. This method had been used for distinguishing the crystal structure in our previous works.(Qibin Li, Meng Wang, Yunpei Liang, Liyang Lin, Tao Fu, Peitang Wei, Tiefeng Peng*. Molecular dynamics simulations of aggregation of copper nanoparticles with different heating rates. Physica E, 2017, 90: 137-142.;Qibin Li, Cheng Huang, Yunpei Liang*, Tao Fu, Tiefeng Peng*. Molecular dynamics simulation of nanoindentation of Cu/Au thin films at different temperatures. Journal of Nanomaterials, 2016, 2016: 9265948.;Qibin Li*, Xianghe Peng, Tiefeng Peng*, Qizhong Tang, Chao Liu, Xiaoyang Shi. Molecular dynamics simulations of coating process: influences of thermostat methods. Journal of Computational and Theoretical Nanoscience, 2016, 13: 4629-4633.;Qibin Li*, Xianghe Peng, Tiefeng Peng*, Qizhong Tang, Xiaomin Zhang, Cheng Huang. Molecular dynamics simulation of Cu/Au thin films under temperature gradient. Applied Surface Science, 2015, 357: 1823-1829.).

The references recommended [47] have been added in the revision.

Point 4: On page 5, please define the parameters in the equation for the radial distribution function.

Response 4: Done. We have specified the parameters in the equation for the radial distribution function (in page 5, line 160-161).

Point 5: Page 7, regarding the surface roughness, the results show that changes in the surface roughness occurs upon sputtering. Is it possible to quantify the degree of order/disorder in the crystal structure before and after sputtering? Order parameters can be of help in this case.

Response 5: Thank you for providing order parameters for quantify the degree order in solid structures. Since the present work focus on the surface effect caused by the incident energy, we would like to discuss the degree of order/disorder in the crystal structure in our future works.

With kind regards,

Sen Tian 

Round 2

Reviewer 1 Report

The ms has improved during revision. It is in acceptable form for publication.

Reviewer 2 Report

If you can refine proof reading of the text that will make it clearer.

Reviewer 3 Report

Manuscript has been improved sufficiently to be published.